# Genome-scale model of metabolism and gene expression provides a multi-scale description of acid stress responses in *Escherichia coli*

**Bin Du**[1], **Laurence Yang**[1], **Colton J. Lloyd**[1], **Xin Fang**[1], **Bernhard O. Palsson**[1,2] *

**1** Department of Bioengineering, University of California, San Diego, La Jolla, California, United States of America, **2** Novo Nordisk Foundation Center for Biosustainability, Technical University of Denmark, Kemitorvet, Kongens, Lyngby, Denmark

* palsson@ucsd.edu

**Data Availability Statement:** All relevant data are within the manuscript and its Supporting Information files.

## Abstract

Response to acid stress is critical for *Escherichia coli* to successfully complete its life-cycle by passing through the stomach to colonize the digestive tract. To develop a fundamental understanding of this response, we established a molecular mechanistic description of acid stress mitigation responses in *E. coli* and integrated them with a genome-scale model of its metabolism and macromolecular expression (ME-model). We considered three known mechanisms of acid stress mitigation: 1) change in membrane lipid fatty acid composition, 2) change in periplasmic protein stability over external pH and periplasmic chaperone protection mechanisms, and 3) change in the activities of membrane proteins. After integrating these mechanisms into an established ME-model, we could simulate their responses in the context of other cellular processes. We validated these simulations using RNA sequencing data obtained from five *E. coli* strains grown under external pH ranging from 5.5 to 7.0. We found: i) that for the differentially expressed genes accounted for in the ME-model, 80% of the upregulated genes were correctly predicted by the ME-model, and ii) that these genes are mainly involved in translation processes (45% of genes), membrane proteins and related processes (18% of genes), amino acid metabolism (12% of genes), and cofactor and prosthetic group biosynthesis (8% of genes). We also demonstrated several intervention strategies on acid tolerance that can be simulated by the ME-model. We thus established a quantitative framework that describes, on a genome-scale, the acid stress mitigation response of *E. coli* that has both scientific and practical uses.

## Author summary

Understanding the acid resistance mechanisms of *E. coli* has important implications in the food, health care and biotechnology industries. The ability of *E. coli* to tolerate acid stress can be attributed to its various regulatory, metabolic and physiological mechanisms. Although different acid resistance mechanisms have been well characterized, few studies are focused on understanding how these mechanisms work together to protect *E. coli* from acid stress. A mathematical representation of the metabolic flux state and proteome

**Funding:** BD, LY, CJL, XF, BOP received Novo Nordisk Foundation Grant NNF10CC1016517 (www.novonordisk.com). BD, LY, CJL, BOP received National Institute of General Medical Sciences of the National Institutes of Health Grant R01GM057089 (www.nigms.nih.gov). The funders had no role in study design, data collection and analysis, decision to publish, or preparation of the manuscript.

**Competing interests:** The authors have declared that no competing interests exist.

allocation of *E. coli* allows the characterization of how these mechanisms interact and function on a systems-level. Here, based on an existing *E. coli* model framework, we characterize three acid stress mitigation responses of *E. coli*: 1) change in membrane lipid fatty acid composition, 2) change in periplasmic protein stability over external pH and periplasmic chaperone protection mechanisms, and 3) change in the membrane protein activities. The predictions of our framework with the integrated mechanisms demonstrated good agreement with RNA sequencing data of *E. coli* on gene expression changes under acid stress. The efforts here open up various opportunities for practical applications, e.g. intervention strategies that challenge acid stress tolerance and enhancement of acid resistance during organic acid production in a cell factory.

## Introduction

Multiple studies have focused on the ability of *Escherichia coli* to tolerate acid stress [1–6]. *E. coli* has been shown to survive under extreme acid stress at pH 2 for several hours and to grow under acid stress above pH 4.5 [1,4–6]. The ability to tolerate acid stress is critical for *E. coli* to complete its life cycle as an enteric bacteria. For colonization in the human digestive tract, it has to pass through the stomach with pH 1.5 to 3, and then metabolize and proliferate at around pH 5 to 6 in the intestinal tract [7,8]. A fundamental understanding of the acid resistance mechanisms of *E. coli* thus has important implications in the food and health care industry, e.g., the development of effective strategies against pathogenic *E. coli* by targeting specific acid resistance mechanisms.

Various acid resistance mechanisms exist that protect *E. coli* under acid stress and are found across different cellular compartments. In the cytoplasm, mechanisms that actively consume protons include four types of amino acid decarboxylase systems and formate hydrogen lyase [9–13]. Metabolism of secondary carbon sources and sugar derivatives are upregulated as these carbon sources produce fewer acids compared to glucose when metabolized [14,15]. Additionally, cytoplasmic buffering from inorganic phosphates, amino acid side chains, polyphosphates, and polyamines helps to maintain intracellular pH homeostasis [16]. When cytoplasmic pH drops under extreme acid stress, cytoplasmic chaperones such as Hsp31 bind and protect unfolded protein intermediates; DNA-binding proteins bind and protect DNA [17–19]. On the inner membrane, activities of electron transport chain components and composition of membrane lipids change under acid stress [14,15,20,21]. In the periplasmic space, periplasmic chaperones HdeA and HdeB are activated under acid stress to bind and protect unfolded protein intermediates [22]. Lastly, outer membrane porins are bound by polyphosphate or cadaverine to reduce proton influx [23,24].

While there have been extensive studies describing the response of *E. coli* under acid stress, research to elucidate how different acid resistance mechanisms function together to protect *E. coli* against a low pH environment is lacking. Such an explanation will require a detailed characterization of different acid resistance mechanisms of *E. coli*. The genome-scale metabolic model (M-model) of *E. coli* provides a mathematical representation of its metabolic capabilities and serves as an ideal framework to describe the acid stress response of *E. coli* [25]. Recently, M-models have been extended to include the synthesis of the gene expression machinery (called ME-models) [26,27]. In addition to computing the optimal metabolic flux state of the organism, ME-models compute the optimal proteome allocation for a given phenotype [27,28], thus providing additional information on the cellular processes as a whole. Furthermore, the calculation on proteome allocation can be validated with RNA sequencing data,

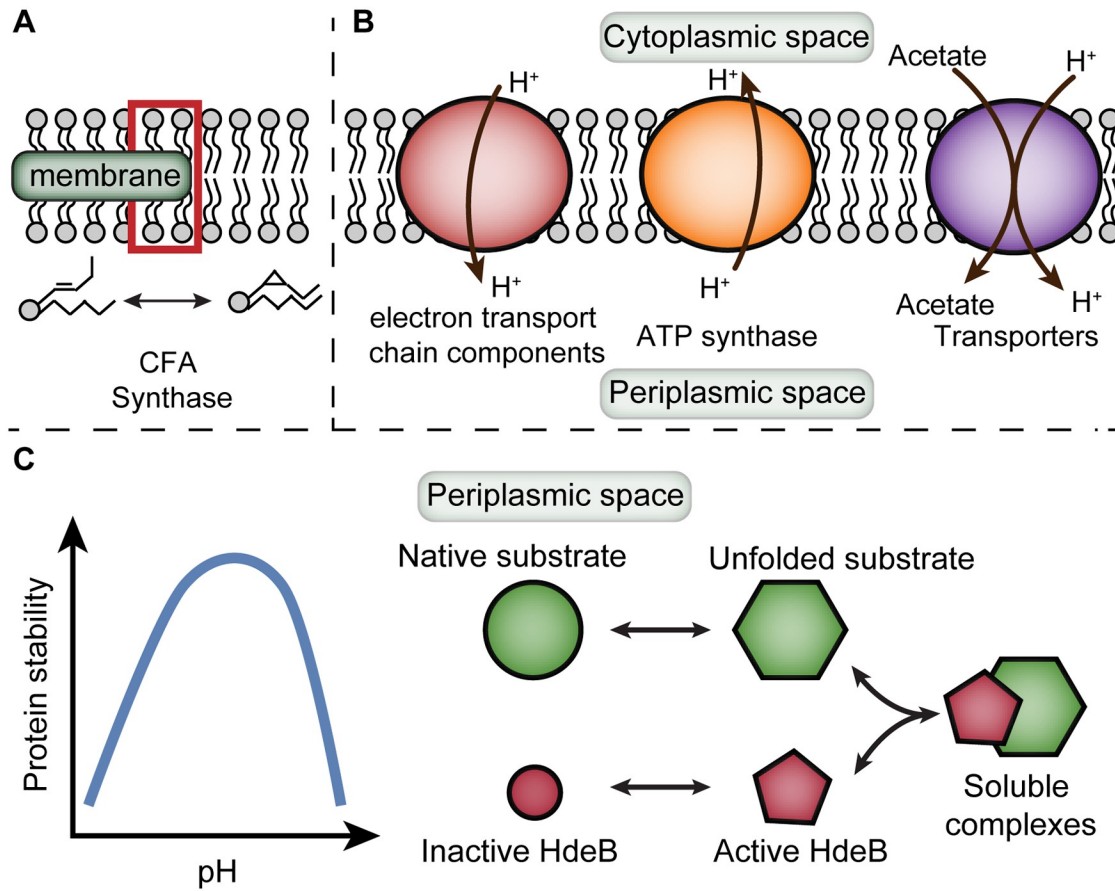

**Fig 1. Illustrations of three different stress response mechanisms of *E. coli* under acid stress.** (A) Adjustment of membrane lipid fatty acid composition. (B) Change in periplasmic protein stability and periplasmic chaperone protection. (C) Activity change of membrane proteins.

which can be conveniently obtained with the advancement of next-generation sequencing technology.

In this work, we characterize the growth of *E. coli* under mild acid stress using the ME-model framework (Fig 1). Mild acid stress can be found under a variety of conditions, including the intestinal tract and fermented food, where the pH is around 5 to 6 [7,29]. We thus narrow the pH range under study between 5.5 and 7, to elucidate the change in cellular responses under acidic and neutral conditions. As a result, we do not include the description of several known acid resistance mechanisms due to the context of acid stress response. One such example is the amino acid decarboxylase systems that are involved in maintaining pH homeostasis. The relevant reactions are included in the ME-model, but do not carry flux in model simulations. The decarboxylase systems are typically active when there is a large influx of proton into the cytoplasmic space or under extreme acid stress when the intracellular pH of *E. coli* drops to around 4 to 5 [13]. Similarly, the descriptions of DNA-binding proteins and the activation of periplasmic chaperone HdeA are not included, but will be more relevant under extreme acid stress [13]. We also do not include some acid resistance mechanisms due to the limitation of the current ME-model framework. One such example is cytoplasmic buffering. The description of this mechanism requires a detailed characterization of the metabolites and amino acid side chains at different protonation states, which is currently out of the model's scope.

Here, we describe three acid stress mitigation mechanisms in the ME-model framework. We first incorporate the change in fatty acid composition of membrane lipids into the ME-model, based on experimental measurements under mild acid stress. Next, we model the change in periplasmic proteins under acid stress, specifically on protein stability and periplasmic chaperone protection. We also model the change in activity for proteins located in the inner membrane of *E. coli*, including ATP synthase, electron transport chain components, and transporters. We integrate all these modifications into the ME-model and compare the simulations with RNA sequencing data of *E. coli* grown under neutral pH and mild acid stress. Specifically, we examine the upregulated and downregulated genes, as well as the change in cellular processes based on cluster of orthologous group (COG) annotation [30]. Lastly, we demonstrate how ME-model can be used to predict intervention strategies on acid tolerance.

## Results

### Adjustment of *E. coli* membrane lipid fatty acid composition under acid stress

The *E. coli* membrane serves as a barrier between the intracellular space and the external environment by controlling the entry and exit of ions and molecules of different sizes. The components of the membrane have been shown to actively respond to changes in the external environment [31]. Specifically, membrane lipids are important components in maintaining membrane function and integrity under environmental perturbations. Several studies have demonstrated that the membrane lipid composition of *E. coli* changes under acid stress, resulting in the change of membrane fluidity that potentially reduces the leakage of protons into the cytoplasm [20,21,32]. Here, we will recapitulate this response in the context of the *E. coli* ME-model framework.

The current ME-model provides a detailed description of the proteins and lipids that constitute the inner and outer membranes of *E. coli* [27]. However, it does not include the constraint that the membrane surface area is completely occupied by proteins and lipids. Therefore, we need to add this constraint into the current ME-model to describe this acid stress response. Our incorporation of the membrane area constraint was able to reproduce the results of similar earlier work (S1 Fig) [33].

Earlier study showed that the composition of fatty acid tails on the membrane lipids of *E. coli* changes during adaptation to acid stress [20]. Specifically, the mole fraction of monounsaturated fatty acids decreased during adaptation, while the proportion in saturated fatty acids and cyclopropane fatty acids increased. This trend is consistently observed across all *E. coli* strains examined by Brown et al [20]. Notably, the composition in cyclopropane fatty acids increased significantly (from an average of 1.57% to 19.6% out of the total fatty acid content) during acid adaptation. We obtained a total of 11 profiles of membrane lipid fatty acid composition of *E. coli* strains from the study by Brown et al. [20] and the existing M-model reconstruction [25]. We grouped the profiles into two categories: the group with an acid-adapted profile where *E. coli* was grown under acidic pH and the group with a non-adapted profile where *E. coli* was grown under neutral pH (Fig 2A).

We incorporated the change in membrane lipid fatty acid composition into the *E. coli* ME-model, while maintaining consistency on the biomass composition and membrane surface area constraints [33,34]. Specifically, the mole fractions of membrane lipids with different fatty acid tails are transformed to their relative fractions in biomass following the procedures in a previous work [34], with units in millimole per gram dry weight of biomass. The calculated lipid biomass fractions are used as the coefficients of lipids in the ME-model reaction on biomass function [27]. The ME-model predicted the group with the acid-adapted profile to have

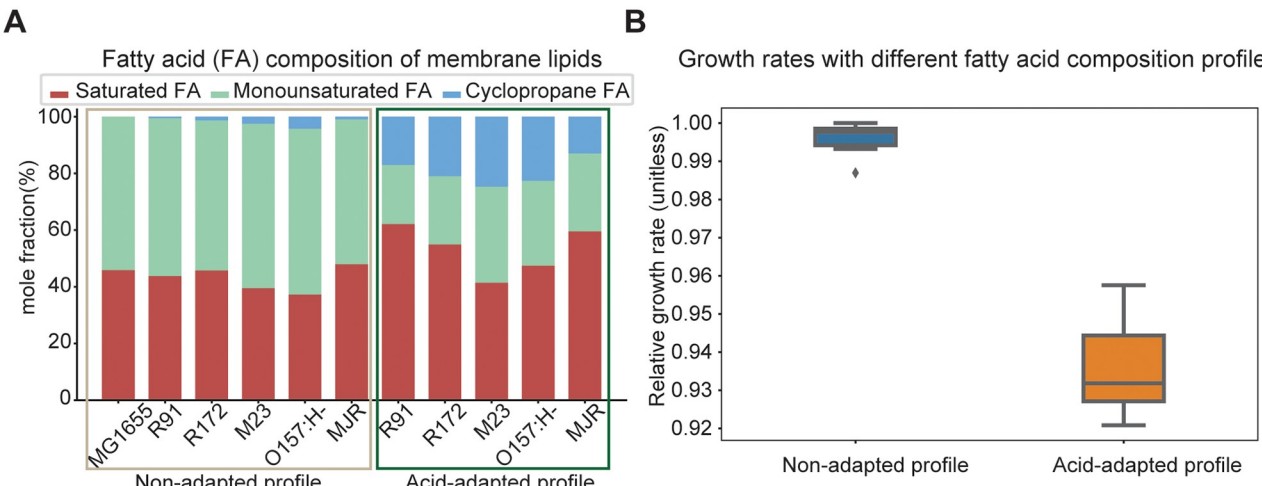

**Fig 2. Fatty acid composition of membrane lipids under different pH conditions.** (A) Comparison of calculated acid-adapted (AA) against non-adapted (NA) fatty acid composition profiles for different *E. coli* strains. The fatty acid composition profiles are calculated based on published data [20]. (B) Comparison of simulated *E. coli* growth rates with different fatty acid composition profiles incorporated into the ME-model. The use of the experimentally determined changes in membrane composition under acid stress leads to around 6% decrease in the computed growth rate.

lower relative growth rates (0.94 ± 0.01) compared to the group with the non-adapted profile (1.00 ± 0.01) (p value $5.93 \times 10^{-6}$) (Fig 2B).

Based on model simulation, we found the *cfa* gene to have the largest change in expression level between the acid-adapted profile and the non-adapted profile. The product of the *cfa* gene, cyclopropane fatty acyl phospholipid synthase, catalyzes the transfer of the methyl group from S-adenosyl-L-methionine (SAM) to convert unsaturated fatty acids to cyclopropane fatty acids. The other genes with the largest computed change in expression levels are mainly associated with the recycling of S-adenosyl-L-methionine and cover a variety of cellular processes including methionine metabolism (*luxS*, *metK*, *metE*), nucleotide metabolism (*purN*, *deoD*), and folate metabolism (*metF*, *folD*) (S1 Table).

## Periplasmic protein stability as a function of pH and periplasmic chaperone protection

Under mild acid stress, *E. coli* maintains intracellular pH within a narrow range (7.4–7.6) [16,35]. However, the pH of the periplasm is close to the external pH when *E. coli* is exposed to an acidic environment [36]. The acidic pH in the periplasm poses a challenge to the periplasmic proteins. *E. coli* has developed strategies to protect periplasmic proteins from acid-induced damage, using molecular chaperones HdeA and HdeB that bind to native substrates to reduce protein denaturation and aggregation [22]. Here, we focus on modeling the change in periplasmic protein stability and the protection by molecular chaperones on periplasmic proteins under acid stress.

Protein stability as a function of pH depends on the $pK_a$s and protonation states of the amino acid side chains of the protein [37–40]. Specifically, protein stability can be described using folding energy ($\Delta G_{folding}$), which is the difference between the folded state and unfolded state of the protein. For the same protein, a more negative folding energy indicates greater stability. An empirical approach has been developed that calculates $\Delta G_{folding}$ based on the number of amino acids of the protein [37,41]. To account for the change in $\Delta G_{folding}$ as a function of

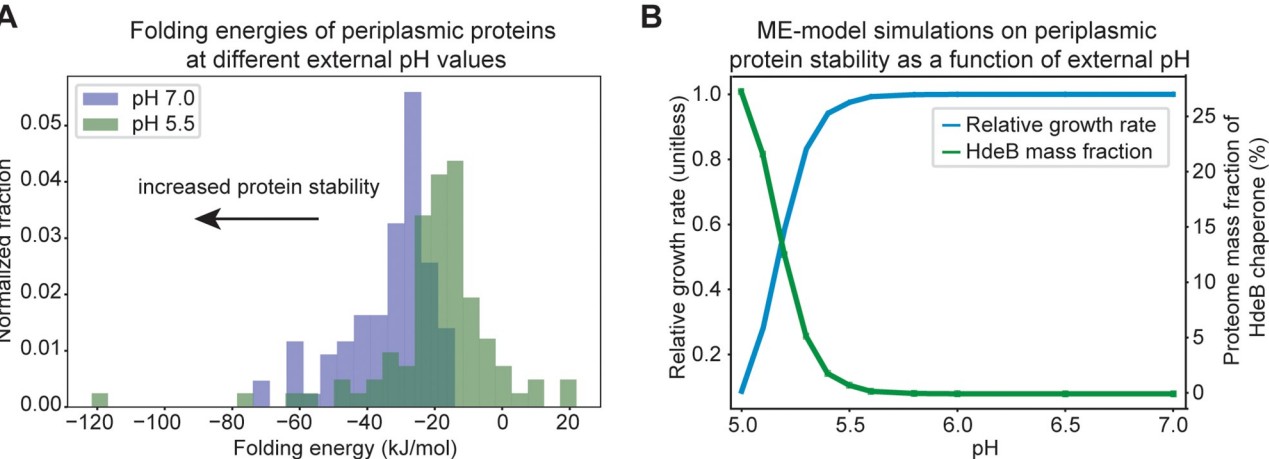

**Fig 3. Periplasmic protein stability is reflected in protein folding energies.** (A) Comparison of calculated folding energies of *E. coli* periplasmic proteins at external pH 7.0 and pH 5.5. Proteins with lower folding energies are generally more stable. Therefore, the periplasmic proteins are found to be more stable at external pH 7 compared to pH 5.5. (B) ME-model simulations on relative growth rate and HdeB mass fraction at different external pH conditions. We calculated the folding energies of periplasmic proteins as a function of pH and modeled the relative ratio of folded and unfolded states of each protein in the ME-model (Materials and methods). We also included the binding of HdeB chaperone to the unfolded states. We then simulated the change in *E. coli* growth rate due to change in protein stability under different external pH conditions. We also showed the change of HdeB mass fraction of the total proteome as a function of external pH.

pH, Ghosh and Dill [37] expressed $\Delta G_{folding}$ as the sum of two terms,

$$\Delta G_{folding} = \Delta G_{neutral} + \Delta G_{electric} \qquad (1)$$

where $\Delta G_{neutral}$ is the energy term that does not consider any charge effect and $\Delta G_{electric}$ accounts for electrostatic interactions and is a function of pH. The term $\Delta G_{electric}$ is protein-specific and depends on the charge and radius of gyration of the folded and unfolded states (Materials and methods).

We calculated the profiles of $\Delta G_{folding}$ as a function of pH for 86 of 93 periplasmic proteins in the ME-model (S2 Table). Folding energies of the other 7 proteins could not be calculated due to issues associated with protein charge calculation (Materials and methods). We also compared the folding energies of the periplasmic proteins under pH 7 and pH 5.5. We found that proteins under pH 7 generally have lower $\Delta G_{folding}$ than those under pH 5.5 (Fig 3A), indicating greater stability for proteins under neutral pH. Notably, all periplasmic proteins examined are favorable towards folding under pH 7 (Fig 3A). We also determined the optimal pH for each protein under study, where $\Delta G_{folding}$ is the lowest and the protein is most stable under the optimal pH. We found that while most proteins have optimal pH around 7, a large number of them have optimal pH around 12 and some have optimal pH around 3 (S2 Fig).

We describe the relationship between the folded and unfolded states of the protein in the form of a ME-model reaction, similar to the approach in the previous work [42]. Specifically, the ratio between the folded and unfolded states of the protein can be calculated from

$$\Delta G_{folding} = -RTln([\text{Folded}]/[\text{Unfolded}]) \qquad (2)$$

where $R$ is the ideal gas constant, $T$ is the temperature, [Folded] and [Unfolded] are the concentrations of the folded and unfolded protein states. The ratio is expressed as the metabolite coefficient in the ME-model reaction, connecting the folded and unfolded states of the protein (Materials and methods). Next, to model periplasmic chaperone protection, we focus on the mechanisms of HdeB, since HdeB has an optimal activation pH from 4 to 5, while HdeA is

most active under pH 2 to 3 [43]. We described HdeB protection on the protein in the form of a ME-model reaction, in which the HdeB protein binds to the unfolded state of the protein to form a chaperone-protein complex (Materials and methods).

Incorporating the description on periplasmic protein stability and HdeB protection in the ME-model, we simulated the response of *E. coli* under different external pH conditions. We found the relative growth rate to decrease slowly as pH decreases from 7 to 5.5, but drops quickly when pH decreases beyond 5.5 (Fig 3B). Similarly, we observed the mass fraction of HdeB of the total proteome to change slowly before pH decreases to 5.5 and increases significantly as pH decreases from 5.5 to 5. We found both the change in protein stability and increase in chaperone synthesis (S3 Fig) contribute to the decrease in growth rate, and the stability change of LptA protein was found to be the major factor causing the decrease in growth rate and increase in HdeB mass fraction. LptA protein is involved in the transport of $(KDO)_2$-lipid $IV_A$, which contributes to *E. coli* biomass [44]. Based on ME-model simulations, genes with the largest change in expression levels as a result of decreasing pH are *hdeB* (periplasmic chaperone), *lptA* (lipopolysaccharide Biosynthesis), *rpoE* (transcription), and *secBDEFGY* (Sec translocation processes) (S3 Table). The Sec complexes are responsible for translocating the LptA protein from the cytoplasm into the periplasmic space [33].

## Membrane protein activity as a function of pH

Under mild acid stress, *E. coli* maintains pH homeostasis in the cytoplasm (around 7.4) while its periplasmic pH is close to that of the external acidic environment [16,35,36]. Thus, the difference in proton concentration across the inner membrane results in a large proton motive force [45,46]. For membrane proteins involved in proton import/export processes, their activities can be significantly affected by the change in proton motive force at different external pH conditions. These proteins include ATP synthase, electron transport chain components, and various membrane transporters. Here, we model the change of their activities, specifically the rates of reactions they catalyze, under mild acid stress and integrate these changes into ME-model simulations.

We first modeled the activity change of ATP synthase under mild acid stress using an existing kinetic model [47]. Specifically, the model consists of a series of elementary reactions that describes the proton transport and the rotation of the rotor subunit in ATP synthase. The rate of ATP synthesis is expressed in terms of the proton concentrations in the cytoplasm and periplasm, as well as the kinetic parameters of the elementary reactions (Materials and methods). It is worth mentioning that ATP synthesis rate also depends on the membrane potential [47,48] and different sets of kinetic parameters are needed when the membrane potential changes under different external pH values. Thus, we fitted the experimental data by Fischer and Gräber [48] on ATP synthesis rate as a function of transmembrane pH difference at three different transmembrane potentials and obtained three parameter sets for rate calculation. The calculated ATP synthesis rates at different external pH values can be found in Fig 4A.

Next, we examined the activity change of the electron transport chain components and various membrane transporters. There is not much evidence available on these reaction mechanisms in terms of the detailed elementary steps. Thus, we modeled their reaction rates based on the theory of nonequilibrium thermodynamics [49]. Specifically, the rate is expressed in terms of the reaction energy, the membrane potential, periplasmic and cytoplasmic proton concentrations, as well as the concentrations of metabolites involved (Materials and methods). The calculations on the electron transport chain components show that the rates of reactions they catalyze remain almost unchanged from neutral pH to acidic pH (S4A Fig). We were unable to calculate the reaction rates for most of the membrane transporters, due to missing

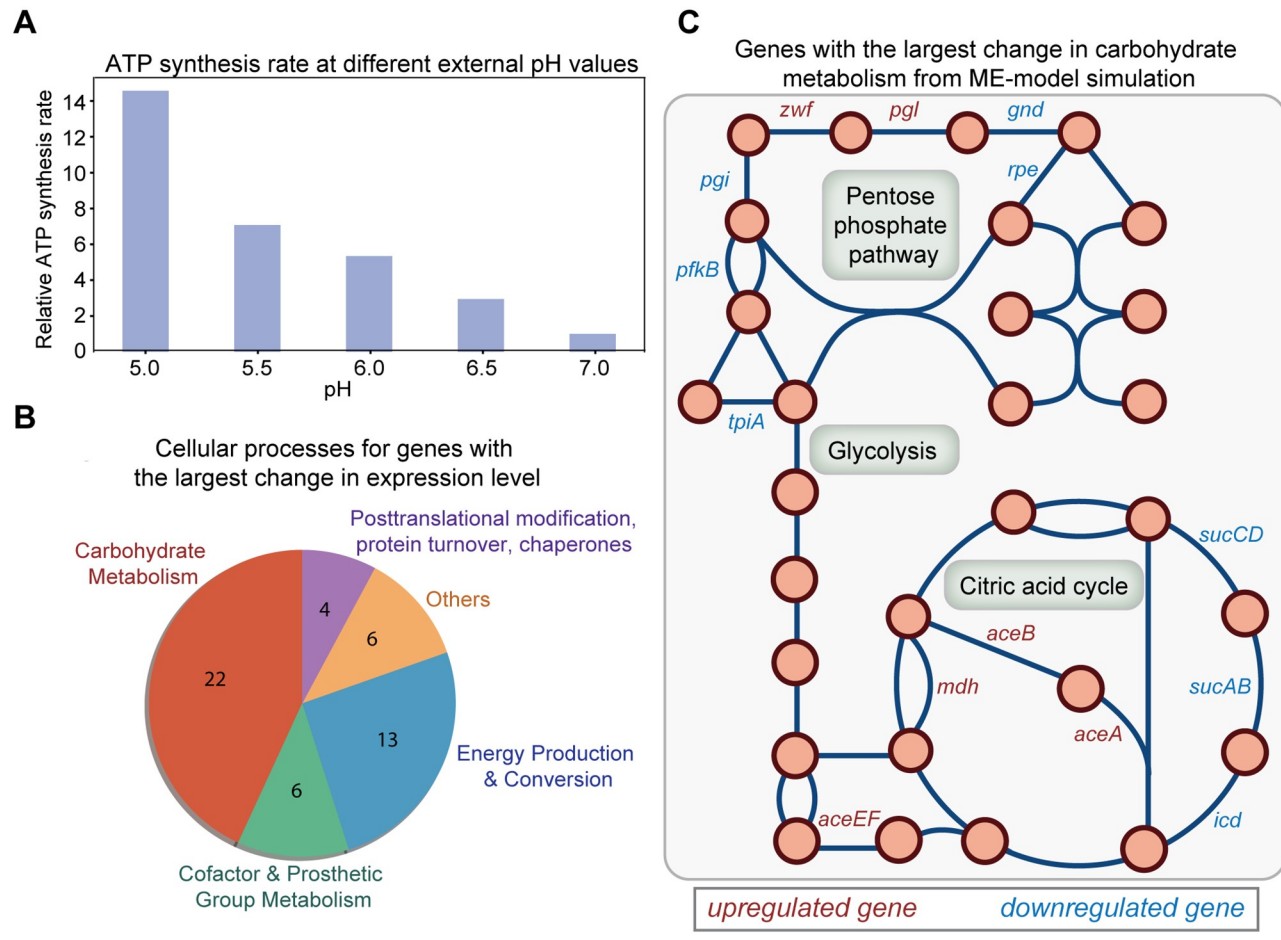

**Fig 4. Change in ATP synthesis rate at different external pH values and the effect on cellular processes simulated using the ME-model.** (A) Relative ATP synthesis rates calculated at different external pH values. The relative ATP synthesis rate at pH 7 is set to 1. (B) Genes with the largest computed change in expression level at pH 5.5 compared to pH 7. The top 51 genes with the largest change in expression level are grouped based on their assignment to the indicated cellular processes. (C) Genes with the largest change in carbohydrate metabolism from ME-model simulations. Most of the genes are involved in three main processes in central metabolism (glycolysis, pentose phosphate pathway and citric acid cycle) and are displayed on the metabolic network map. Genes predicted to be upregulated are colored in red and genes predicted to be downregulated are colored in blue.

metabolite concentration data. However, we found that the change of their activities in terms of reaction rates had minimal impact on cellular growth rate and processes (<1%) through the sensitivity analysis using the ME-model (S4B Fig).

Based on the analysis on the activity change of different membrane proteins across pH, we modeled the change of ATP synthesis rate at different external pH values by modifying the effective turnover rate ($k_{eff}$) of the reaction catalyzed by ATP synthase in the ME-model [27] (Materials and methods). Considering possible errors due to parameter fitting, we performed sensitivity analysis and found the change in cellular processes at different ATP synthesis rates to be similar. Using the calculated ATP synthesis rate at pH 5.5 as an example (Fig 4A), the top 50 genes with the largest change in expression levels are mainly involved in carbohydrate metabolism (e.g., citric acid cycle, glycolysis/gluconeogenesis, pentose phosphate pathway) and energy production and conversion (oxidative phosphorylation related to ATP synthase) (Fig 4B, S4 Table). The top reactions with the largest increase in ATP use under pH 5.5 cover processes including ATP maintenance requirement, glycolysis/gluconeogenesis, nucleotide

salvage pathway, amino acid metabolism, purine and pyrimidine biosynthesis, translation process, etc (S5 Table).

## ME-model with integrated mechanisms explains the acid stress response of *E. coli*

We integrated the description of the three pH stress mitigation mechanisms (membrane lipid fatty acid composition, periplasmic protein stability and periplasmic chaperone protection, and the activity change of membrane proteins) into the ME-model, and then simulated its response under neutral pH and mild acid stress (pH 5.5). We compared the simulations to RNA sequencing data of K-12 MG1655 *E. coli* strains grown under pH 7 and pH 5.5 in glucose minimal medium from a previous study [50]. The *E. coli* strains from which the RNA-seq data were obtained include: 1) the wild type strain, 2) two strains adapted to pH 5.5 through adaptive laboratory evolution [51], and 3) two control strains adapted to specific media conditions. Since the acid-adapted strains were evolved in glucose minimal medium with lowered magnesium concentration and MES buffer, the two control strains (one for lowered magnesium concentration and one for MES) were necessary to account for the possible effects due to these two changes in media composition.

We compared RNA-seq data (S6a–S6e Table) and ME-model simulations (S6f Table) in terms of the differentially expressed genes (DEGs) due to acid stress (growth under pH 5.5 versus pH 7). We grouped the DEGs found in RNA-seq data into three categories (S6g Table): 1) DEGs currently not active in the ME-model, 2) DEGs correctly predicted by the ME-model, and 3) DEGs incorrectly predicted by the ME-model.

We found a large number of genes in the first category to be associated with membrane proteins and transporters and their related cellular processes (S6h Table). For example, one of the reported acid stress responses involves the blockage of outer membrane porins by secreted cadaverine [52]. Therefore, these DEGs are currently outside the ME-model's predictive capabilities. To include such descriptions in the ME-model, quantitative measurements on cadaverine binding to outer membrane porin and the corresponding change under acid stress is required.

For genes in the second category, we found that, on average, 80% of the upregulated genes in the RNA-seq data to be correctly predicted. These correctly predicted DEGs are mainly involved in the translation process (45% of genes), membrane proteins and related processes (18% of genes), amino acid metabolism (12% of genes), and cofactor and prosthetic group biosynthesis (8% of genes). Additionally, we found a limited number of downregulated genes to be active in the ME-model, as shown in Fig 5A. For genes in the third category, those found to be upregulated in the data but predicted to be downregulated in the ME-model are grouped by COG categories and shown in Fig 5B. Genes found to be downregulated in the data but predicted to be upregulated by the ME-model are discussed in more detail below.

We grouped the correctly predicted DEGs by COG categories (Materials and methods) and summarized them by the underlying mechanisms in Fig 5A. We found a large number of upregulated genes to be related to the translation process (Fig 5B red). We found the increase in ATP synthase activity as the main driver for these upregulated genes, as a large number of them were also upregulated when simulating with only modified ATP synthase activity in the ME-model (S4 Table). Additionally, we found increased expression levels for a number of proteins on the inner and outer membranes of *E. coli*. These proteins include the electron transport chain components, transporters (uptake of sugar, lysophospholipid), Sec translocase, and BAM complex responsible for outer membrane assembly. We also found cofactor and prosthetic group biosynthesis to be another major category with a number of upregulated genes.

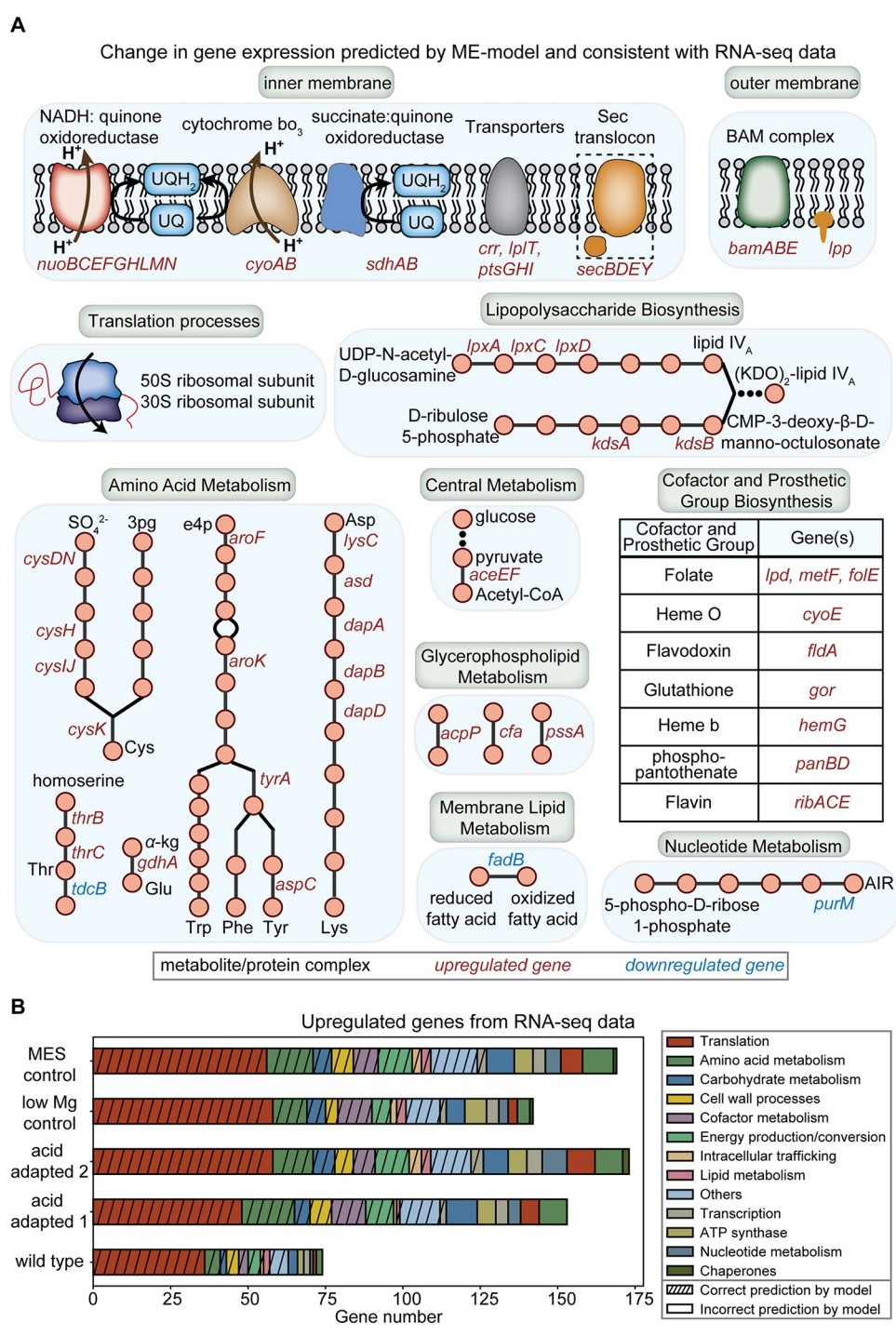

**Fig 5. Comparison of ME-model simulations, accounting for the three acid stress mechanisms, against RNA-seq data from *E. coli*.** (A) Differentially expressed genes (DEGs) due to acid stress found to be consistent with model predictions and RNA-seq data. We grouped the list of DEGs found into different COG categories. (B) Upregulated genes in RNA-seq data compared to ME-model simulations. The upregulated genes in RNA-seq data correctly predicted by the ME-model (with strikes in different COG categories) and incorrectly predicted by the ME-model (no strikes in different COG categories) are shown. We listed the five *E. coli* strains with RNA sequencing data available. The two control strains are labeled as "low Mg control" to account for the effect of lowered magnesium concentration and "MES control" to account for the use of MES buffer during adaptive evolution of *E. coli* under acid stress.

The differential expression of certain genes in this category (e.g. *metF*, *folE*, *fldA*) were related to the recycling of cofactor SAM, due to the upregulation of *cfa* to adjust the membrane lipid fatty acid composition under acid stress. Due to changes in membrane lipid fatty acid composition and periplasmic proteome predicted by the ME-model, we found upregulated genes in RNA-seq data to be related to membrane lipid metabolism, lipopolysaccharide biosynthesis, and glycerophospholipid metabolism.

Furthermore, the correctly predicted DEGs in amino acid metabolism cover processes related to cysteine, threonine, lysine, glutamate, and aromatic amino acids (tryptophan, tyrosine, phenylalanine). We found the upregulated genes in amino acid metabolism to be mainly due to the requirement of the related metabolic processes as well as proteome resources for various cellular processes. First, for metabolic processes, the upregulation in cysteine biosynthesis contributes to the biosynthesis of methionine, which is an important metabolite involved in the recycling of SAM. The increased biosynthesis of threonine mainly contributes to the biosynthesis of glutamate. The increased biosynthesis of glutamate contributes to several metabolic processes: 1) biosynthesis of lysine through aspartate (later used for protein synthesis), 2) biosynthesis of asparagine (later used for protein synthesis), 3) recycling of SAM, as glutamate leads to methionine and tetrahydrofolate, which are intermediates of the SAM recycling process). Next, we examined the processes that have the largest change in proteome resource requirement under acid stress. For each amino acid in Fig 5A, we obtained the top 100 translation reactions in the ME-model that has the largest change in flux under acid stress, respectively. We then obtained the overlapping translation reactions from those amino acids. We examined the specific proteome sectors that correspond to the protein product of these translation reactions. We found that processes such as oxidative phosphorylation, cofactor and prosthetic group biosynthesis, glycolysis/gluconeogenesis to be the main categories requiring additional proteome resources under acid stress (S7 Table), and hence are the major drivers behind the upregulation of amino acid biosynthesis shown in Fig 5A.

We then examined the incorrectly predicted DEGs. We found a few genes to be downregulated in the RNA-seq data but predicted to be upregulated (*rlmC*, *glcD*, *hisI*, *erpA*, *nadB*). Upon examining the reactions catalyzed by these gene products, we found proton generation to be involved in three reactions, with the corresponding genes being *rlmC*, *hisI*, *nadB*. The proton generation in the reaction explains the downregulation of these genes, as *E. coli* tends to minimize proton production under acid stress. Genes found to be upregulated in the data but downregulated in ME-model predictions were grouped based on the COG categories (Fig 5B). These incorrectly predicted DEGs suggest ways to further develop the modeling of acid stress response. For example, the arginine-dependent acid resistance system has been shown to play a role under acid stress [11], but the corresponding genes were not correctly predicted by the ME-model. A possible way to improve model predictions is to fine-tune model parameters related to arginine metabolism based on RNA-seq data. We also found genes related to cytoplasmic chaperones to be upregulated in RNA-seq data but not predicted by the ME-model. A previous reconstruction of the cytoplasmic chaperone network in the ME-model exists [42] and its incorporation can potentially improve predictions of the use of chaperone related processes.

In addition, we examined the data on protein abundance under pH 6 and pH 7 by Schmidt et al [53]. We obtained the list of proteins that were differentially expressed under acid stress (S8 Table, confidence score > 500). Comparing the list against the differentially expressed genes predicted by the ME-model, we found that the expression change of 123 proteins were correctly predicted and 119 proteins were incorrectly predicted (S8 Table). We found the largest number of proteins to belong to the translation process, for both the correctly predicted and incorrectly predicted lists. The model predicted most of the upregulated genes but few

downregulated genes in the translation process correctly. The result indicates that the model is able to predict certain processes that are more active under acid stress after incorporating various acid stress response mechanisms. The list of incorrectly predicted translation genes can potentially suggest downregulated processes under acid stress to include in the model.

### ME-model simulates intervention strategies on *E. coli* acid tolerance

To demonstrate the use cases of the ME-model, we examined several potential intervention strategies on *E. coli* acid tolerance using the developed framework. The intervention strategies are designed based on the acid stress responses of *E. coli* described in this work. Specifically, they are downregulation of the HdeB protein and knockout of the *cfa* gene.

As demonstrated earlier, HdeB chaperone plays an important role in protecting unfolded proteins in the periplasm. Since we have HdeB as the only periplasmic chaperone here, knockout of the hdeB gene would result in no growth predicted by the model. We thus simulated the ME-model with 10% of the expected HdeB protein amount under pH 5.5. We found that with reduced HdeB protein expression, the growth rate drops to 31.6% of that with normal HdeB expression under pH 5.5. Genes with the largest change in expression include *ilvBHIN* (amino acid metabolism), *lysS* (tRNA charging), *rpe* (pentose phosphate pathway), *lysU*, *deoA*, *udp*, *tdk*, *yjjG* (nucleotide salvage pathway) (S9 Table). The results here can be used to compare with experimental studies examining downregulation or knockout of the *hdeB* gene. Any discrepancies between the experimental outcome and model simulations can potentially lead to the discovery of novel chaperone protection mechanisms in the periplasm [54].

We also assessed the effect of *cfa* gene knockout using the ME-model. *E. coli* with the *cfa* gene knocked out cannot convert unsaturated fatty acids to cyclopropane fatty acids. Thus, the membrane fluidity of *E. coli* under acidic conditions is likely to be similar to that under neutral condition, while increased proton gradient under acidic conditions increases the proton leakage into the cytoplasm [20,21,32]. We simulated the ME-model with *cfa* gene knockout and proton influx into the cytoplasm at 10 mmol • $gDW^{-1}$ • $hr^{-1}$. We have examined the results at various proton influx rates and found the qualitative trend of gene expression change to be the same. We found that the largest change in gene expression cover processes such as amino acid metabolism (*glnA*, *argBC*), carbohydrate metabolism (*paaH*, *idnK*), oxidative phosphorylation (*atpADE*), membrane and transport related processes (*lpp*, *ompN*, *hcaT*) (S10 Table). The experimental validation on this intervention strategy is straightforward, by knocking out the *cfa* gene and characterizing the gene expression profiles through RNA sequencing data. Any discrepancies between experimental and modeling results can help uncover new strategies that *E. coli* uses to adjust the membrane lipid fatty acid composition under acid stress.

## Discussion

In this study, we described the response of *E. coli* under acid stress using the ME-model framework. We first modified the membrane lipid fatty acid composition based on experimental data, with the addition of the constraint on total membrane surface area. Second, we modeled the pH-dependent periplasmic protein stability and periplasmic chaperone protection mechanisms. Third, we characterized the activities of membrane proteins under low pH. Lastly, we integrated these descriptions of stress mitigation mechanisms into the ME-model and compared the simulations of the integrated model with measured RNA sequencing data and proteomics data. We demonstrated that the ME-model was able to recapitulate DEGs under acid stress in a number of cellular processes, including amino acid metabolism, cofactor and prosthetic group biosynthesis, processes related to membrane proteins, and translation process.

The effects of acid stress mitigation on these cellular processes can now be understood at the systems level and quantitatively computed. We also suggested a few areas for further model development, based on model predictions that were inconsistent with the RNA-seq data. We also demonstrated several use cases of the developed ME-model, by proposing intervention strategies on acid tolerance that can be validated experimentally.

The work here describes the change in the cellular state of *E. coli* between two distinct conditions, the mild acidic condition and the neutral condition. A continuous profile of the change in cellular processes as the pH decreases from neutral to acidic can provide more insights into how *E. coli* adjusts its cellular resource allocation when facing increased acid stress. However, such an effort is currently limited due to the lack of relevant experimental data. For example, the current data on fatty acid composition of membrane lipids of *E. coli* are only measured under pH 5 and 7. Possible steps forward include acquiring more experimental data at the intermediate pH values between 5 and 7 or making simplifying assumptions about how the fatty acid composition profile changes over pH.

ME-model simulations predicted only a few of the periplasmic proteins to be active. The main reason for the inactivation of other proteins is the lack of description of their downstream processes or metabolic reactions they catalyze in the ME-model. The addition of relevant processes could help provide a more complete picture of the periplasmic protein response under acid stress, as the stability profiles for most of the periplasmic proteins are available from this work. Furthermore, adding these descriptions can uncover more periplasmic proteins that significantly affect the growth rate and cellular processes, and potentially improve the predictions on acid stress response.

The ME-model framework here enables predictions of how different interventions affect the acid stress tolerance of *E. coli*. For example, we can design intervention strategies on the recycling of S-adenosyl-L-methionine, which is an important cofactor responsible for the adjustment of membrane lipid fatty acid composition under acid stress. As another example, the effect of *hdeB* knockout can be simulated using the ME-model and compared with experimental data. Discrepancies between model simulations and the data can potentially lead to discoveries of novel periplasmic chaperone protection mechanisms [54].

Taken together, the work here describes acid stress mitigation responses in *E. coli* through a mechanistic approach and provides insights into the resulting changes to its cellular processes. It is worth noting that the current description focuses on the acid stress response of *E. coli* under the aerobic growth condition with glucose as the sole carbon source. In practice, *E. coli* faces more complicated nutrient environments and can be subjected to anaerobic respiration. The response to acid stress differs due to different environmental conditions (e.g., activation of formate hydrogen lyase under anaerobic acid stress [13]). Thus, descriptions of additional acid resistance mechanisms can be added to expand the scope of ME-model predictions. The study here is a first step towards a complete characterization of the wide array of acid stress responses of *E. coli*.

## Materials and methods

### ME-model and simulations

The ME-model framework is based on the work by Lloyd et al [27], with no change on the parameters used other than the inclusion of acid stress mitigation responses described in the text. A quad-precision NLP solver was used to obtain the ME-model solutions [55]. The source code for model construction and integration of the acid stress mitigation mechanisms is available on GitHub (https://github.com/bdu91/acidify-ME). All work here is in implemented in Python 2.7.6.

## Stability of periplasmic proteins as a function of pH

As mentioned in the main text, protein stability can be quantified by the folding energy $\Delta G_{folding}$, which is the sum of $\Delta G_{netural}$ and $\Delta G_{electric}$ based on Eq 1. The change in pH affects the value of $\Delta G_{electric}$, which can be expressed as

$$\Delta G_{electric} = kT\left(\frac{Q_{folded}^2\, l_b}{2R_{folded}(1 + \kappa R_{folded})} - \frac{Q_{unfolded}^2\, l_b}{2R_{unfolded}(1 + \kappa R_{unfolded})}\right) \tag{3}$$

where $Q_{folded}$ and $Q_{unfolded}$ are the protein charges in the folded and unfolded states, $R_{folded}$ and $R_{unfolded}$ are radius of gyration of the folded and unfolded states, $k$ is the Boltzmann constant, $T$ is the temperature, $l_b$ is the Bjerrum length and $\kappa = 2c\, l_b$ ($c$ as the salt concentration, set as 0.25 M here) [37].

The charge of the unfolded state of the given protein can be calculated based on the p$K_a$s and charges of the individual amino acid side chains (S11 Table). The charge of the folded state can be obtained through a method called multi-conformation continuum electrostatics (MCCE), which calculates the p$K_a$s and charges of the amino acid side chains of the folded state [56]. The MCCE method requires the PDB structures of the folded proteins, which were obtained from the latest genome-scale metabolic network reconstruction of *E. coli* [25]. It is worth mentioning that the charge of 7 periplasmic proteins cannot be calculated due to failed delphi runs in the MCCE method. The radius of gyration of the folded protein $R_{folded}$ is calculated through the Bio3d package in R [57], using the PDB structure of the folded protein. The radius of gyration of the unfolded protein $R_{unfolded}$ is obtained by fitting empirical data (S12 Table) and the relationship between the number of amino acid residues $N$ and $R_{unfolded}$, where $R_{unfolded} \propto N^{0.588}$ [58].

Finally, as $\Delta G_{folding}$ at neutral pH can be calculated based on the number of amino acids of the protein [37,41], $\Delta G_{folding}$ at different pH values can be obtained from the change of $\Delta G_{electric}$ over pH using Eq 3.

## Periplasmic chaperone protection by HdeB in the ME-model

We first modeled the formation of HdeB protein, including steps on transcription, translation, translocation from the cytoplasm to the periplasm and formation of HdeB dimer [43]. The details of each step have been defined in the COBRAme framework by Lloyd et al [27]. We then modeled the protection of HdeB on unfolded proteins. We defined a spontaneous folding reaction for each periplasmic protein, using the coupling constraint defined by Ke et al [42]. Specifically, we have

$$K[\text{HdeB}] + (1 + K + \mu/k_{folding})[\text{Unfolded}] \leftrightarrow K[\text{HdeB} - \text{unfolded} - \text{complex}] + [\text{Folded}] \tag{4}$$

where [HdeB] is the HdeB protein, [Unfolded] and [Folded] are the folded and unfolded states of the protein, [HdeB − unfolded − complex] is the complex formed by HdeB bound to the unfolded state, $K$ is the ratio between the unfolded state and the folded state and can be obtained from $\Delta G_{folding}$ under the given pH, $\mu$ is the growth rate in the ME-model, $k_{folding}$ is the kinetic folding rate and can be calculated based on the work by Gromiha et al [59]. For proteins where $\Delta G_{folding}$ cannot be obtained, we assume the protein is favorable towards folding under all conditions and set $\Delta G_{folding}$ to -100 kJ/mol.

There is no data available on the amount of HdeB under different pH conditions. Instead, we let the model to produce enough HdeB to bind and protect the proteins in the unfolded state. Based on the steady state assumption, the amount of HdeB required under the specific pH condition is determined by the amount of unfolded proteins through mass balance. Thus,

we have $[HdeB] = \sum_{p=1}^{n} \left[ \text{Unfolded} \right]_p \left( \frac{1 + K_p + \mu_p / k_{folding-p}}{K_p} \right)$, where there exists n different proteins in the periplasm.

## Activity of ATP synthesis rate as a function of external pH in the ME-model

We used the kinetic model by Jain and Nath [47] to describe the mechanism of ATP synthase through a list of elementary steps, including proton transport and rotor rotation. The rate of ATP synthesis can be expressed in terms of the cytoplasmic and periplasmic proton concentrations, as well as the kinetic parameters.

$$v = k_1 / (1 + k_2 * H^+_{cytoplasm} / H^+_{periplasm} + k_3 / H^+_{periplasm}) \tag{5}$$

It is worth mentioning that parameters $k_1$, $k_2$ and $k_3$ are composite terms. Each term consists of various kinetic parameters of the elementary steps.

We used the experimental data from Fischer and Gräber [48], where the rate of *E. coli* ATP synthase was measured as a function of transmembrane pH difference at three different transmembrane potentials (80 mV, 108 mV, 152 mV). Based on Eq 5, we obtained three sets of kinetic parameters at different membrane potentials by fitting the experimental data through a non-linear least-squares minimization procedure [60].

To calculate the rate of ATP synthesis under a specific external pH, we first calculated the cytoplasmic pH, using the relationship between the cytoplasmic pH and the external pH derived by Slonczewski et al [35]. We next calculated the membrane potential of *E. coli* under the given external pH based on the experimental measurements by Felle et al [61]. From the three fitted parameter sets at different membrane potentials, we selected the set with the closest membrane potential. Using the selected parameter set and the calculated pH values, we calculated the rate of ATP synthesis under different external pH conditions. To standardize the calculated rates, we defined the rate under pH 7 as 1 and expressed the rates under other pH values as the fold change relative to it. To incorporate the change in ATP synthesis rate under the specific external pH in the ME-model, we adjusted the effective turnover rate ($k_{eff}$) of ATP synthase in the ME-model according to the calculated fold change under the given external pH [27].

## Activity of electron transport chain components as a function of pH

For electron transport chain components, we examined those active in the ME-model simulations, which are NADH dehydrogenase (associated with ubiquinone-8), and cytochrome oxidase bo3. We described the rate as a function of pH using the derivation by Jin and Bethke [49], based on the theory of nonequilibrium thermodynamics. Specifically, the rate is expressed as,

$$v = v_+ \left( 1 - exp \left( \frac{-nF\Delta E^\circ + mF\Delta\psi}{RT} \right) * \left( \frac{[H^+_{periplasm}]^m [D^+]^{v_{D+}} [A^-]^{v_{A-}}}{[H^+_{cytoplasm}]^m [D]^{v_D} [A]^{v_A}} \right) \right) \tag{6}$$

where $v_+$ is the forward reaction flux, $n$ is the number of electrons transferred, $\Delta E^\circ$ is the difference in standard redox potential between the donating and accepting half-reactions, $m$ is the number of protons transported across the membrane, $\Delta\psi$ is the membrane potential, $F$ is Faraday's constant, $R$ is the ideal gas constant, $T$ is the temperature, $[D^+]$ and $[D]$ are the concentrations of the oxidized and reduced forms of the electron-donating half reaction, $[A]$ and $[A^-]$ are the concentrations of the oxidized and reduced form of the electron-accepting half reaction. Since we were only interested in the relative change of activity for the electron transport

chain components, we focused on calculating the term after $v_+$ in Eq 6. The difference in standard redox potential as termed $\Delta E°$ is calculated based on the standard redox potential of the half-reactions from multiple sources [62–64]. The membrane potential $\Delta\psi$ at the specific external pH is calculated based on the experimental measurements by Felle et al [61]. The concentrations of the electron donors and acceptors are obtained from the experimental measurements by Bennett et al [65].

## Comparison of DEGs between ME-model predictions and RNA sequencing data

We computed the amount of individual proteins expressed in the ME-model and determined the relative change of each protein expression from neutral pH to acidic pH. We compared the change in protein expression to the DEGs in the RNA sequencing data in terms of the direction of change. For a more systematic comparison of DEGs, we grouped the *E. coli* genes into cellular processes based on COG annotation (detailed list in S13 Table). Different *E. coli* strains have different sets of DEGs under acid stress in the RNA-seq data, with a small set of DEGs overlapping. Thus, we compared the DEGs found in each strain against the DEGs predicted by the ME-model and grouped the correctly and incorrectly predicted DEGs by COG categories. To obtain the set of genes consistent between model predictions and RNA-seq data, we obtained the list of COG categories commonly found across all five *E. coli* strains in which the correctly predicted genes fall. For each COG category, we then summarized the list of correctly predicted genes from all five *E. coli* strains.

## Supporting information

**S1 Fig. The change in relative growth rate as a function of fraction of membrane surface area covered by proteins.** We examined the change in protein fraction in both the inner and outer membranes. Similar results in an earlier version of the ME-model can be found in the work by Liu et al [33] (Fig 5A). It is worth mentioning that the qualitative trend in terms of the change of growth rate matches with the earlier work, but discrepancies in the quantitative change of growth rate exist. Such discrepancies are mainly due to the change in membrane composition description in the latest version of the ME-model [27], which was used as the framework in this study.
(TIF)

**S2 Fig. Optimal pH of *E. coli* periplasmic proteins.** The optimal pH of the specific protein is determined based on the pH where the folding energy is the lowest.
(TIF)

**S3 Fig. The relative growth rates across pH under two scenarios: 1) by taking into account only the increased HdeB synthesis under acidic conditions, 2) by taking into account both the change in protein stability and increased HdeB synthesis under acidic conditions.** We found that when considering the the change in protein stability under acidic conditions, the growth rate dropped significantly, compared to when only considering the increased HdeB synthesis.
(TIF)

**S4 Fig.** (A) Change in enzyme activity of electron transport chain (ETC) components as a function of pH. Here we focused on the ETC components active in the ME-model and calculated the change of their activity at different external pH values based on the theory of non-equilibrium thermodynamics (main text Materials and methods). We found that the two

electron transport chain components examined does not have a notable change in enzyme activity across pH. (B) Change in growth rate due to change in the activities of membrane transporters. We focused on the membrane transporters that are active in the ME-model simulations. We change the activity of the membrane transporters one at a time and simulated the corresponding growth rates. We found that the change in the activities of membrane transporters do not significantly affect the growth rate (stayed at 1.0 relative growth rate). (TIF)

**S1 Table. Top genes with the largest change in expression with the adjustment of membrane lipid fatty acid composition under acidic conditions.**
(XLSX)

**S2 Table. Folding energies of periplasmic proteins across pH from 0 to 14.**
(XLSX)

**S3 Table. Top genes with the largest change in expression considering periplasmic protein stability and periplasmic chaperone protection mechanisms under acid stress.**
(XLSX)

**S4 Table. Top genes with the largest change in expression with the increase in ATP synthase activity (10 fold increase).**
(XLSX)

**S5 Table. Top reactions that consume ATP upon the increase in ATP synthase activity.**
(XLSX)

**S6 Table. Differentially expressed genes found in RNA-seq data of five *E. coli* strains (S2a– S2e Table) and predicted by model (S2f Table).**
(XLSX)

**S7 Table. Top translation reactions requiring proteome resources that lead to the upregulation of amino acid synthesis in Fig 5A.**
(XLSX)

**S8 Table. Validation of model predictions against the proteomics data of *E. coli* under pH 6 compared to pH 7.**
(XLSX)

**S9 Table. Top 100 genes with the largest change in expression with 10% of HdeB activity compared to 100% HdeB activity under pH 5.5.**
(XLSX)

**S10 Table. Top 100 genes with the largest change in expression with increased proton influx (at 10 mmol • gDW$^{-1}$ • hr$^{-1}$) and *cfa* gene knockout compared to wild type normal condition.**
(XLSX)

**S11 Table. p$K_a$ values of amino acids.**
(XLSX)

**S12 Table. Data on the radius of gyration of unfolded proteins.**
(XLSX)

**S13 Table. COG categories.**
(XLSX)

## Acknowledgments

We thank Ke Chen for the valuable discussions on the *E. coli* periplasmic protein stability and chaperone protection mechanisms.

## Author Contributions

**Conceptualization:** Bin Du, Bernhard O. Palsson.

**Data curation:** Bin Du.

**Formal analysis:** Bin Du.

**Funding acquisition:** Bernhard O. Palsson.

**Investigation:** Bin Du, Laurence Yang.

**Methodology:** Bin Du, Colton J. Lloyd.

**Project administration:** Bin Du, Bernhard O. Palsson.

**Resources:** Bin Du, Bernhard O. Palsson.

**Software:** Bin Du, Colton J. Lloyd.

**Supervision:** Laurence Yang, Bernhard O. Palsson.

**Validation:** Bin Du.

**Visualization:** Bin Du, Xin Fang.

**Writing – original draft:** Bin Du.

**Writing – review & editing:** Bin Du, Laurence Yang, Xin Fang, Bernhard O. Palsson.

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
