## [Decision Letter · Decision Letter 0]

20 Aug 2019

Dear Dr Palsson,

Thank you very much for submitting your manuscript 'Genome-scale model of metabolism and gene expression provides a multi-scale description of acid stress responses in Escherichia coli' for review by PLOS Computational Biology. Your manuscript has been fully evaluated by the PLOS Computational Biology editorial team and in this case also by independent peer reviewers. The reviewers appreciated the attention to an important problem, but raised some substantial concerns about the manuscript as it currently stands. While your manuscript cannot be accepted in its present form, we are willing to consider a revised version in which the issues raised by the reviewers have been adequately addressed. We cannot, of course, promise publication at that time.

Sincerely,

Anders Wallqvist

Associate Editor

PLOS Computational Biology

Daniel Beard

Deputy Editor

PLOS Computational Biology

[LINK]

Reviewer's Responses to Questions

**Comments to the Authors:**

Reviewer #1: Based on an existing framework, this study characterizes the three acid stress mitigation responses of E. coli: 1) change in membrane lipid fatty acid composition, 2) change in periplasmic protein stability over external pH and periplasmic chaperone protection mechanisms, and 3) change in the activities of membrane proteins. The predictions by the ME model with the three integrated mechanisms demonstrated good agreement with RNA sequencing data of E. coli on gene expression changes under acid stress. It is valuable to mathematically reproduce genome-wide changes, but I have several comments.

1) It is hard to understand how the authors integrate chaperone protection mechanisms. How did they estimate the HdeB amount necessary for folding acid-induced unfolded proteins at different pHs? Did they estimate or measure the total protein amounts in periplasm? While Eq.4 does not show any rate equations, how did they estimate the synthesis flux of HdeB at different pHs?

2) More explanation on the ME model is required. Especially, how does it estimate macromolecule or chaperone synthesis flux? Does it consider transcriptional regulations (e.g. sigma factors)? Because readers like to know how chaperone synthesis regulation quantitatively affects the genome-scale metabolism.

3) Did the authors evaluate or measure decreased activities of membrane proteins (e.g. transporters)? Membrane protein activity would decrease because the periplasm sides of the proteins are exposed to a low pH.

4) The simulated cell growth decreased under acid stress. Is it caused by increased chaperone synthesis? Are there any other reasons for the decreased growth?

5) In Fig 4A, the ATP synthesis rate was very high at pH 5. Can they explain how the produced ATP is used within a cell?

6) The ATP synthesis flux (Fig 4) is critically important in this study, thus it seems necessary to experimentally measure the ATP synthesis flux.

7) In Fig 5B, “Others” are correctly predicted by the ME model. However, “others” are meaningless. Can they specify the gene names or their functions?

Reviewer #2: The authors extend their genome-scale model of E. coli metabolism to account for the effects of mildly acidic extracellular conditions. The model development is well explain and the comparisons to published data are promising. I have no major concerns about the study, but I do believe that the manuscript might be improved if the following comments are addressed:

1. The author summary reads too much like the abstract and should be rewritten from a broader perspective.

2. The five E. coli strains used should be mentioned earlier in the paper, like line 129.

3. The authors should reconcile the statement on page 18 "We found the rates of electron transport chain components to be almost unchanged (<1%) from neutral pH to acidic pH. We were unable to calculate the reaction rates for most of the membrane transporters, due to missing metabolite concentration data. However, we found that the change of their activities had minimal impact on cellular growth rate and processes (<1%) through the sensitivity analysis using the ME-model" with the comment on page 20 "Additionally, we found upregulated expression for a number of proteins on the inner and outer membranes of E. coli. These proteins include the electron transport chain components,...".

4. Is there some reason that the DEGs were not analyzed for false negatives (i.e. the model produces not change, but the genes are differentially expressed in experiment)? See lines 316-319.

5. The reason for limiting the modeled pH range from 5.5-7.0 should be mentioned earlier in the paper. See lines 416-431.

Reviewer #3: In the present manuscript, Du et al aim to get a systems-level understanding of the acid stress response of E. coli. Towards this end, the authors incorporate three known acid resistance mechanisms into a previously developed genome-scale metabolic and expression (ME) model of E. coli. In a first, step, they consider these mechanisms (namely altered fatty acid composition in membranes, periplasmatic protein stabilization by chaperones, and changes in ATP synthase/ETC protein activity) separately and show that their model recapitulates previously reported behavior. For example, the authors predict that an acid-adapted fatty acid profile is accompanied by a reduction in growth rate, as is also observed experimentally. Finally, the authors combine all three acid resistance mechanisms to predict the response of their ME model to mild acid stress. As a metric of predictive power, the authors compare predicted and measured changes in gene expression, and show for 80% of the genes predicted to be upregulated in acid stress the experimental data shows a consistent upregulation.

Overall, the manuscript tackles an interesting biological problem, and presents the model implementation of different acid resistance mechanisms in a very clear fashion. However, several points in the final part of the manuscript were unclear to me:

1) The stated goal of this manuscript is to provide a “fundamental understanding” (lines 18-19) of the acid stress response. However, I felt that the manuscript struggles to provide an actual systems-level understanding of this response. For example, the authors report numerous expression changes in amino acid metabolism and translation genes, which are not obvious (at least to me). Can the authors provide a rationale for these changes? Are these a consequence of e.g. massive overexpression of HdeB (which according to the authors accounts for up to 25% of the protein mass at pH 5)? Currently, the manuscript relies too heavily on merely describing the (predicted or observed) changes in expression, without necessarily providing insights on why these changes might occur.

2) It was unclear to me why the authors focus on upregulated genes (i.e. Figure 5B, lines 329-330). Given that the ME model includes proteome allocation, I would expect some parts of the proteome to shrink as well, right? In general, I would like to see a table with the actual model predictions and the experimental RNAseq data in the supplementary material of the final manuscript.

3) Ultimately, in its current form this manuscript is a well-executed consistency check: based on what we know about the molecular mechanisms of acid stress resistance, do we predict a cellular response to acid stress that is consistent with experimental data? But as the authors state themselves (lines 433-434), a key asset of such a model is its ability to design intervention strategies to reduce acid tolerance. However, the authors currently refrain from predicting intervention strategies (barring the short discussion in lines 434-438), which is a pity. I would suggest that the authors include a results section, in which they examine at least some potential intervention strategies, with a focus on those interventions that are also experimentally feasible (even if the authors decide not to test any of these interventions themselves).

Additional points:

1) Figure 1S: it was unclear to me what exact earlier results are being reproduced in this figure. Maybe add some more explanation in the figure caption, or include these previous results in the figure?

2) I was intrigued by the model predicting that a change in FA composition is enough to reduce growth rate by ~10% (Figure 2B). Do the authors have an explanation for this? Along the same lines, the authors predict that Cfa has the largest change in expression in the acid-adapted profile. Would overexpression of Cfa be sufficient to a) change lipid composition and b) reduce growth rate?

3) In lines 155 to 162 the authors refer to expression changes in various genes, without providing a pointer to a supplementary table of figure. Please make sure to include such a pointer. The same accounts for the text in lines 231 to 242.

4) Lines 201-202: “…indicating that protein stability might be an underlying factor that influences the protein expression in the periplasm.” This sentence was unclear to me: do the authors suggest that designated regulatory mechanisms that expression the respective proteins in a pH-dependent manner?

5) The authors identify LptA as “the major factor causing the drop in growth rate and increase in HdeB mass fraction” (lines 236-237). Do the authors predict that overexpression of LptA, or expression of an LptA protein variant from a related bacterial species with better lower pH tolerance, would alleviate the growth defect? If yes, are data on such a validation experiment available in the literature?

6) To validate their predictions, the authors compare gene expression with RNA-seq data. I’m curious whether the authors can also use published proteomics data (i.e. from a data set in which E.coli was subjected to pH 6, PMID 26641532) for validation?

7) Can the authors include the numbers of DEGs for the different categories mentioned in lines 316 – 318?

8) The authors should include the full list of DEGs in their final model (basis for Figure 5) in the supplementary material.

**Have all data underlying the figures and results presented in the manuscript been provided?**

Reviewer #1: Yes

Reviewer #2: Yes

Reviewer #3: No: As described above, there are several instances in which the authors highlight example results, without providing a pointer to the full results/analysis.

PLOS authors have the option to publish the peer review history of their article (what does this mean?). If published, this will include your full peer review and any attached files.

Reviewer #1: No

Reviewer #2: No

Reviewer #3: No

---

## [Decision Letter · Decision Letter 1]

1 Nov 2019

Dear Dr Palsson,

We are pleased to inform you that your manuscript 'Genome-scale model of metabolism and gene expression provides a multi-scale description of acid stress responses in Escherichia coli' has been provisionally accepted for publication in PLOS Computational Biology.

In the meantime, please log into Editorial Manager at https://www.editorialmanager.com/pcompbiol/, click the "Update My Information" link at the top of the page, and update your user information to ensure an efficient production and billing process.

One of the goals of PLOS is to make science accessible to educators and the public. PLOS staff issue occasional press releases and make early versions of PLOS Computational Biology articles available to science writers and journalists. PLOS staff also collaborate with Communication and Public Information Offices and would be happy to work with the relevant people at your institution or funding agency. If your institution or funding agency is interested in promoting your findings, please ask them to coordinate their releases with PLOS (contact ploscompbiol@plos.org).

Thank you again for supporting Open Access publishing. We look forward to publishing your paper in PLOS Computational Biology.

Sincerely,

Anders Wallqvist

Associate Editor

PLOS Computational Biology

Daniel Beard

Deputy Editor

PLOS Computational Biology

Reviewer's Responses to Questions

**Comments to the Authors:**

Reviewer #1: It is improved.

Reviewer #2: I am satisfied with the responses and revisions offered by the authors.

Reviewer #3: The revised manuscript is a clear improvement over the initial submission, and the changes made by the authors were very helpful in clarifying their work to me. All of my concerns have been adequately addressed, and I am happy to recommend this work for publication.

**Have all data underlying the figures and results presented in the manuscript been provided?**

Reviewer #1: Yes

Reviewer #2: Yes

Reviewer #3: Yes

PLOS authors have the option to publish the peer review history of their article (what does this mean?). If published, this will include your full peer review and any attached files.

Reviewer #1: No

Reviewer #2: No

Reviewer #3: No

---

## [Editor Report · Acceptance letter]

15 Nov 2019

PCOMPBIOL-D-19-00947R1 

Genome-scale model of metabolism and gene expression provides a multi-scale description of acid stress responses in Escherichia coli

Dear Dr Palsson,

I am pleased to inform you that your manuscript has been formally accepted for publication in PLOS Computational Biology. Your manuscript is now with our production department and you will be notified of the publication date in due course.

With kind regards,

Bailey Hanna
